# CURE4Rec: A Benchmark for Recommendation Unlearning with Deeper Influence

**Chaochao Chen**[1], **Jiaming Zhang**[1], **Yizhao Zhang**[1], **Li Zhang**[1], **Lingjuan Lyu**[2],
**Yuyuan Li**[3,1]*, **Biao Gong**[3], **Chenggang Yan**[3]
[1]Zhejiang University, [2]Sony AI, [3]Hangzhou Dianzi University

{zjuccc, 22321350, 22221337}@zju.edu.cn, zhanglizl80@gmail.com, lingjuan.lv@sony.com
y2li@hdu.edu.cn, a.biao.gong@gmail.com, cgyan@hdu.edu.cn

## Abstract

With increasing privacy concerns in artificial intelligence, regulations have mandated the *right to be forgotten*, granting individuals the right to withdraw their data from models. Machine unlearning has emerged as a potential solution to enable selective forgetting in models, particularly in recommender systems where historical data contains sensitive user information. Despite recent advances in recommendation unlearning, evaluating unlearning methods comprehensively remains challenging due to the absence of a unified evaluation framework and overlooked aspects of deeper influence, e.g., fairness. To address these gaps, we propose CURE4Rec, the first comprehensive benchmark for recommendation unlearning evaluation. CURE4Rec covers four aspects, i.e., unlearning Completeness, recommendation Utility, unleaRning efficiency, and recommendation fairnEss, under three data selection strategies, i.e., core data, edge data, and random data. Specifically, we consider the deeper influence of unlearning on recommendation fairness and robustness towards data with varying impact levels. We construct multiple datasets with CURE4Rec evaluation and conduct extensive experiments on existing recommendation unlearning methods. Our code is released at `https://github.com/xiye7lai/CURE4Rec`.

## 1 Introduction

Over the past few years, growing concerns over information abundance and data leakage have intensified the focus on privacy preservation within artificial intelligence. Regulations such as the General Data Protection Regulation (GDPR) (Union, 2018), the California Consumer Privacy Act (Pardau, 2018)and the Delete Act (Information, 2023) grant individuals the *right to be forgotten*, requiring the deletion of personal data used in information systems. Nowadays, the ubiquitous application of machine learning models in information systems poses potential risks for memorizing training data (Fredrikson et al., 2015). Consequently, the aforementioned regulations also require forgetting the associated data memory within the trained models, giving rise to the concept of machine unlearning. Recently, machine unlearning has gained increasing popularity in computer vision (Bourtoule et al., 2021; Gupta et al., 2021), natural language processing (Chen & Yang, 2023; Eldan & Russinovich, 2023), and recommender systems (Chen et al., 2022; Li et al., 2023a,b). As recommender systems typically rely on historical interaction data to extract user preferences, the recommendation model inherently contains sensitive user information. Therefore, there is a crucial need for unlearning to preserve privacy. The task of machine unlearning in recommender systems is termed as recommendation unlearning.

---

*Corresponding author

38th Conference on Neural Information Processing Systems (NeurIPS 2024) Track on Datasets and Benchmarks.

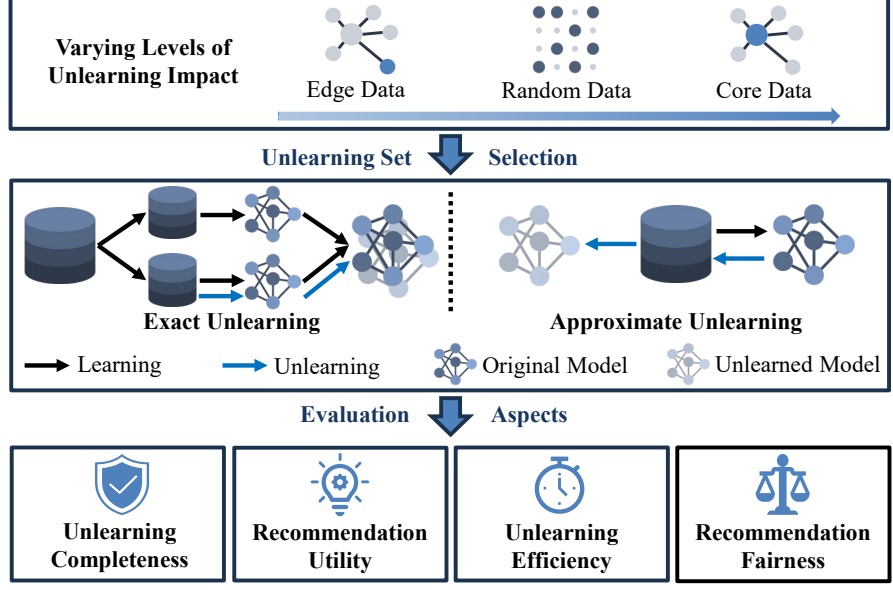

Figure 1: An illustration of CURE4Rec, a comprehensive benchmark tailored for evaluating recommendation unlearning methods. CURE4Rec evaluates unlearning methods using data with varying levels of unlearning impact on four aspects, i.e., unlearning completeness, recommendation utility, unlearning efficiency, and recommendation fairness.

While machine unlearning has demonstrated significant potential in preserving user privacy, conducting a comprehensive evaluation of unlearning methods continues to pose difficulties. Various unlearning methods employ distinct evaluation metrics, yet a universally applicable evaluation framework remains absent. Specifically, existing evaluation methods predominantly focus on the unlearning completeness, unlearning efficiency, and its impact on model utility, overlooking the deeper influence of model properties.

In this paper, we identify two overlooked aspects of deeper influence. *Firstly*, fairness is a crucial consideration for recommendations (Wang et al., 2023), but is often neglected in unlearning evaluations. Ensuring fair recommendation outcomes can avoid user discrimination and enrich the recommendation platform's understanding of user preferences. Existing studies demonstrate that unlearning can affect the fairness of models (Oesterling et al., 2024). *Secondly*, existing evaluation methods neglect the influence of various unlearning sets, randomly selecting data for unlearning. Distinct unlearning sets, however, can result in significantly different impacts on model performance (Fan et al., 2024). Performing comprehensive evaluations on different unlearning data contributes to understanding the robustness of unlearning methods.

To address these issues, we introduce CURE4Rec, a comprehensive benchmark specifically designed to evaluate recommendation unlearning methods. As shown in Figure 1, CURE4Rec's evaluation encompasses four aspects, i.e., unlearning completeness, recommendation utility, unlearning efficiency, and recommendation fairness. Additionally, each aspect is investigated with three data selection strategies, i.e., core data, edge data, and random data. This triadic breakdown tests to reflect the robustness of recommendation unlearning methods towards different unlearning sets. The main contributions of this work are summarized as follows:

- We introduce CURE4Rec, a comprehensive benchmark tailored for evaluating recommendation unlearning methods. CURE4Rec enables evaluation across multiple aspects, including unlearning completeness, recommendation utility, unlearning efficiency, and recommendation fairness.

- To the best of our knowledge, we are the first to investigate the impact of unlearning on recommendation fairness, introducing fairness evaluation to comprehensively grasp its impact and proposing additional requirements to consider for further research.

- We further examine the impact of different unlearning sets. Based on the level of collaboration, we select core data, edge data, and random data to construct unlearning sets respectively, aiming to thoroughly explore the impact towards unlearning completeness, recommendation utility, unlearning efficiency, and recommendation fairness.

- We offer multiple datasets tailored for evaluation using our CURE4Rec. Furthermore, we conduct extensive experiments across existing recommendation unlearning methods and report their performance (please refer to Figure 2 for an overview of our results).

## 2 Related Work

### 2.1 Machine Unlearning

Machine unlearning aims to eliminate the memory of specific data, serving purposes such as privacy protection (Liu et al., 2022) and erasing data biases (Sattigeri et al., 2022; Chen et al., 2024b). According to the level of unlearning completeness, existing machine unlearning methods can be categorized into two approaches, i.e., exact unlearning and approximate unlearning.

*Exact Unlearning (EU)* aims to completely eliminate the influence of target data on the model. The most straightforward method of exact unlearning is retraining the model from scratch on the updated dataset (removing the target data), but this method incurs a significant computational time cost. To mitigate this cost, existing EU methods revamp the training process via ensemble learning, which limits the retraining cost to sub-datasets or sub-models (Bourtoule et al., 2021; Yan et al., 2022).

*Approximate Unlearning (AU)* achieves unlearning through direct parameter manipulation, avoiding the significant time cost of retraining. Most AU methods utilize gradients or influence function to estimate the influence of target data and subsequently remove it from models (Sekhari et al., 2021; Wu et al., 2022; Mehta et al., 2022). Alternatively, other methods directly prune or dampen model parameters to achieve unlearning (Wang et al., 2022; Foster et al., 2024).

### 2.2 Recommendation Unlearning

Recommendation unlearning aims to eliminate the influence of target data within the recommender system (Li et al., 2024b). A naive approach to achieve recommendation unlearning is through the direct application of the classic unlearning method, i.e., SISA (Bourtoule et al., 2021). Due to the collaborative characteristics of recommendation data, tailored methods have been proposed to improve SISA for recommendation unlearning, e.g., RecEraser (Chen et al., 2022) and UltraRE (Li et al., 2023a). In addition to EU methods mentioned above, AU method also enters the scene, utilizing refined influence functions to enable recommendation unlearning (Li et al., 2023b; Zhang et al., 2023). Note that this paper focuses on investigating the model-agnostic approaches. Other approaches focus on specific models, e.g., sequential recommendation (Ye & Lu, 2023), session-based recommendation (Xin et al., 2024), and large language model-based recommendation (Wang et al., 2024; Hu et al., 2024). Note that this paper focuses solely on recommendation unlearning of training data. An alternative line of research, known as attribute unlearning, explores the unlearning of latent user attributes in recommender systems. (Ganhör et al., 2022; Li et al., 2023c; Chen et al., 2024a).

### 2.3 Machine Unlearning Benchmarks

Emerging research has pioneered early investigation into unlearning benchmarks, focusing on image classification (Choi & Na, 2023), large language models (Maini et al., 2024; Li et al., 2024a; Jin et al., 2024), and diffusion models (Zhang et al., 2024). By proposing new datasets or modifying existing ones, these investigations design depth evaluation metrics within their corresponding domains. However, these benchmarks leave unexplored deeper influence of unlearning on model properties, i.e., fairness and robustness. This exploration is crucial for recommender systems, as alternations in the performance of recommendation models immediately affect recommendation lists, eventually influencing use experience. To the best of our knowledge, we are the first to introduce a recommendation unlearning benchmark, and comprehensively explore the deeper influence of unlearning on recommendation fairness and robustness.

# 3 CURE4Rec

In this section, we first recall the process of recommendation unlearning, outlining the necessary inputs for evaluations. Then, we introduce evaluation aspects of our proposed CURE4Rec, detailing specific metrics for each aspect. Finally, we present the strategy for unlearning set selection.

## 3.1 Recommendation Unlearning

The entire process of recommendation unlearning consists of four stages: I) completing learning process to generate the original model; II) determining the unlearning set, i.e., the unlearning target, which is a subset of training data; III) conducting unlearning process based on the original model to produce the unlearned model; and IV) evaluating the unlearned model. To ensure reliable evaluation, we evaluate unlearning methods using identical training and testing data, employing the same learning process to generate the same original model. This ensures that all unlearning methods start from the same baseline in stage I. To investigate unlearning robustness, we select three types of unlearning sets in stage II (Section 3.3). In stage IV, CURE4Rec's evaluation includes the four aspects (Section 3.2).

In the context of recommendation, unlearning targets may vary among users, items, and user-item interactions. Commonly, recommendation unlearning scenarios focus on user-wise unlearning (Li et al., 2023a). Thus, our benchmark primarily investigates the user-wise unlearning scenarios.

## 3.2 Evaluation Aspects

**Unlearning Completeness.**    Unlearning completeness stands as the primary goal and fundamental requirement of recommendation unlearning. Exact unlearning methods inherently guarantee unlearning completeness by retraining, which is the only authorized approach (Thudi et al., 2022). On the other hand, approximate unlearning methods, lacking the ability to achieve authorized unlearning, often require the demonstration of unlearning completeness through theoretical proofs or empirical studies. Therefore, following the completeness evaluation of approximate unlearning in previous studies (Graves et al., 2021; Ma et al., 2022; Li et al., 2023b; Kurmanji et al., 2024), we evaluate unlearning completeness of recommendation unlearning based on the attacking performance of Membership Inference Oracle (MIO).

MIO follows the standard membership inference procedure to evaluate unlearning completeness in image classification task (Graves et al., 2021; Ma et al., 2022). In the context of recommendation, we concatenate user embeddings with the average item embeddings of their respective interacted items as the data features, and the probability of being in the training set as the data label. Please refer to Section 4.5 for more training details. To evaluate unlearning completeness, we query MIO with the unlearned data points. Ideally, MIO outputs 1 (indicating presence in the training set) for the original model and 0 (indicating absence from the training set) for the unlearned model. Since exact unlearning methods guarantee complete unlearning, we only evaluate the completeness of approximate unlearning methods.

**Recommendation Utility.**    Recommendation unlearning aims to erase the memory of target data within recommender systems without causing harm to the knowledge acquired from the remaining data. Thus, preserving the recommendation utility of the remaining data is another important goal of unlearning. To investigate the impact of unlearning on model utility, we employ two widely used metrics, i.e., Normalized Discounted Cumulative Gain (NDCG) and Hit Ratio (HR), to evaluate the recommendation performance of the unlearned model on the testing set. For both metrics, we truncate the ranked list to 20 items.

**Unlearning Efficiency.**    Retraining from scratch represents the gold standard in unlearning, but its practical implementation carries a prohibitive computational overhead. Recommender systems encompass hundreds of thousands of users, generating a large amount of unlearning requests. Therefore, improving unlearning efficiency is a crucial goal of recommendation unlearning. We measure unlearning efficiency by the total runtime of the entire unlearning process, i.e., stage III. Note that we enable parallel training for exact unlearning.

**Recommendation Fairness.**    Previous research has demonstrated that unlearning affects deeper model properties such as fairness (Oesterling et al., 2024). Mitigating the negative impact of

unlearning is also an important requirement of unlearning. In this paper, we evaluate the performance fairness of recommendation unlearning from the following two perspectives: i) the fairness between active and inactive groups (A-IGF), and ii) the fairness among different shards (shardGF), as exact unlearning methods divide the datasets into multiple shards.

For A-IGF, we follow the representative user-oriented group fairness research in recommendation (Li et al., 2021). Based on the number of interactions, we classify the top 5% of users as the active group and the remaining 95% users as the inactive group. Active and inactive users are selected outside the unlearning set, because we aim to investigate the impact on the remainder users. Then we compute the difference of the average recommendation utility, i.e., NDCG@20, between active and inactive groups to represent A-IGF. For shardGF, we report the variance of recommendation utility among all shards to compare the shard-level fairness (Rastegarpanah et al., 2019). Note that we do not compute shardGF for approximate unlearning, because these methods do not involve sharding.

### 3.3 Unlearning Set Selection

Existing evaluation methods typically select data randomly for the unlearning set. However, previous studies have shown that i) poisoned data can be constructed to make it hard to unlearn (Marchant et al., 2022), and ii) different data points have varying difficulty of unlearning (Fan et al., 2024). Motivated by these findings, in this paper, we explore the impact of using varying unlearning sets, which can also reflect the robustness of unlearning.

To significantly demonstrate this impact, we adopt a model-agnostic selection strategy to create three types of unlearning sets: core data (which impacts many other data points), edge data (with minimal impact on others), and random data. Specifically, we regard the user-item interactions as a non-weighted bipartite graph, where users and items are represented as nodes, and an edge connects them if there is an interaction. Existing research suggests that a node's importance correlates strongly with its centrality in a graph (Haveliwala, 2002; Li et al., 2012; Park et al., 2019). In the context of recommendation, centrality is associated with collaborations, manifested as neighbors in a graph. Thus, we define the importance of a node $x$ as follows:

$$I(x) = c(x) \cdot \frac{\sum_{y \in N(x)} c(y)}{|N(x)|}, \tag{1}$$

where $c(x)$ denotes the centrality of node $x$, and $N(x)$ denotes the number of neighbors of node $x$. Due to the collaborative characteristic of recommendation data, we use the degree of node, i.e., the number of first-order neighbors, to compute centrality. Finally, we rank all nodes based on $I(x)$ to select the core data and edge data.

## 4 Experimental Setup

### 4.1 Datasets

We conduct experiments on three real-world datasets widely used in recommendation. **ML-100K**[2]: The MovieLens dataset is one of the most extensively utilized datasets in recommender system research. MovieLens 100k contains 100 thousand individual ratings. **ML-1M**: MovieLens 1M contains 1 million ratings. **ADM**[3]: The Amazon dataset comprises multiple subsets categorized according to different types of Amazon products. One of these subsets, known as the Amazon Digital Music (ADM) dataset, includes ratings of digital music. Following the widely used pre-processing procedure (He et al., 2017; Wang et al., 2019; He et al., 2020), we convert ratings into implicit feedback. The statistical details of these datasets are summarized in Table 1. To avoid extreme sparsity, we filter out the users and items that have less than 5 interactions. For each dataset, we randomly select 80% ratings as the training set, 10% ratings as the validation set, and the remaining as the test set. The unlearning ratio, i.e., the percentage of unlearning set within the training set, is initially set as 5%. We also explore this ratio within a range of (5%, 10%, 15%, 20%) in Appendix A.4.

---

[2]https://grouplens.org/datasets/movielens/
[3]http://jmcauley.ucsd.edu/data/amazon/

Table 1: Summary of datasets.

| Dataset | User # | Item # | Interactions # | Sparsity |
|---------|--------|--------|----------------|----------|
| ML-100K | 943 | 1,349 | 99,287 | 92.195% |
| ML-1M | 6,040 | 3,416 | 999,611 | 95.155% |
| ADM | 478,235 | 266,414 | 836,006 | 99.999% |

## 4.2 Recommendation Models

Aligning with existing studies on recommendation unlearning (Chen et al., 2022; Li et al., 2023b,a), we use three representative recommendation models based on collaborative filtering for evaluation:

- **WMF**: Weighted Matrix Factorization(WMF) (Chen et al., 2020) is a non-sampling recommendation model that treats all missing interactions as negative interactions and assigns them with uniform weights.

- **BPR**: Bayesian Personalized Ranking (Rendle et al., 2012) is a widely used recommendation model that uses a Bayesian personalized ranking objective function to optimize matrix factorization.

- **LightGCN**: LightGCN (He et al., 2020) is the state-of-the-art collaborative filtering model, which improves recommendation performance by simplifying graph neural networks.

## 4.3 Unlearning Methods

We consider the following recommendation unlearning methods, including both EU and AU approaches (note that we set the number of shards to 10 for EU and explore other values in Section 5.5):

- **Retrain**: Retraining from scratch is the goal standard unlearning method.

- **SISA**: SISA (Bourtoule et al., 2021) stands as the classic algorithm for machine unlearning, adaptable to various scenarios, including recommender systems.

- **RecEraser**: RecEraser (Chen et al., 2022) is specifically designed for recommendation unlearning, which modifies SISA to boost performance in recommendation tasks.

- **UltraRE**: UltraRE (Li et al., 2023a) enhances RecEraser for recommendation tasks by modifying two key stages, i.e., division and aggregation.

- **SCIF**: SCIF (Li et al., 2023b) is the first approximate unlearning method in recommendation systems, employing influence functions tailored for recommendation tasks.

## 4.4 Parameters Settings

In the training phase of original models, we randomly sample 4 negative items for each observed interaction following (He et al., 2017). In the case of model-specific hyper-parameters, we tune them in the ranges suggested by their original papers. In detail, the batch size is set to 512, the learning rate is set to 0.01, the embedding size is set to 32. The maximum number of epochs is set to 500. The early stopping strategy is adopted in our experiments, which terminates the training when NDCG@20 on the validation set does not increase for 5 successive epochs.

## 4.5 MIO Training Details

Following (Li et al., 2023b), we adopt an ideal concept, i.e., Membership Inference Oracle (MIO), to evaluate unlearning completeness. Specifically, We implement an approximated MIO via a basic three-layer (64, 16, 4) neural network with ReLu and Softmax as activation functions for hidden layers and the output layer respectively. We train the MIO via stochastic gradient descent with 100 epochs and a learning rate of 0.001. The MIO outputs the probability of the queried data point being in the training set. To evaluate the unlearning completeness, we query MIO with the unlearned data points. Ideally, MIO outputs 1 (being in the training set) for the original model while outputs 0 (not being in the training set) for the unlearned model.

Table 2: Results in terms of unlearning completeness (MIO accuracy - approaching 0.5), recommendation utility (NDCG and HR ↑), and recommendation fairness (A-IGF - approaching Retrain) for the approximate recommendation unlearning method, where `Learn` denotes the results before unlearning. Core, random, and edge respectively refer to the selection of the unlearning sets as core data, random data, and edge data.

| | | ML-100K | | | | ML-1M | | | | ADM | | | |
| --- | --- | --- | --- | --- | --- | --- | --- | --- | --- | --- | --- | --- | --- |
| | | NDCG@20 | HR@20 | MIO | A-IGF | NDCG@20 | HR@20 | MIO | A-IGF | NDCG@20 | HR@20 | MIO | A-IGF |
| `Learn` | | 0.3215 | 0.3415 | 0.722 | -0.0450 | 0.2144 | 0.2112 | 0.741 | -0.042 | 0.0277 | 0.0578 | 0.756 | 0.0167 |
| Retrain | Core | 0.3187 | 0.3295 | 0.540 | -0.0184 | 0.2196 | 0.2174 | 0.544 | -0.0188 | 0.0221 | 0.0446 | 0.555 | 0.0053 |
| | Random | 0.2872 | 0.3353 | 0.538 | -0.0403 | 0.2124 | 0.2108 | 0.547 | -0.0507 | 0.0252 | 0.0519 | 0.556 | 0.0141 |
| | Edge | 0.3091 | 0.3140 | 0.536 | -0.0430 | 0.2148 | 0.2051 | 0.546 | -0.0518 | 0.0272 | 0.0554 | 0.556 | 0.0164 |
| SCIF | Core | 0.2483 | 0.2382 | 0.561 | -0.0322 | 0.1865 | 0.1629 | 0.569 | -0.0213 | 0.0194 | 0.0398 | 0.571 | 0.0094 |
| | Random | 0.2699 | 0.2617 | 0.563 | -0.0268 | 0.1922 | 0.1785 | 0.571 | -0.0311 | 0.0227 | 0.0461 | 0.575 | 0.0106 |
| | Edge | 0.2894 | 0.3012 | 0.601 | -0.0375 | 0.2031 | 0.1811 | 0.623 | -0.0191 | 0.0245 | 0.0502 | 0.579 | 0.0103 |

## 4.6 Hardware Information

We run all experiments on the same Ubuntu 20.04 LTS System server with 48-core CPU, 256GB RAM and NVIDIA GeForce RTX 3090 GPU.

## 5 Results and Discussion

In this section, we report and analyze the results regarding four evaluation aspects under three selections of unlearning sets. We present a visualized overview of compared recommendation unlearning methods in Figure 2. We observe that apart from unlearning completeness, the AU method (SCIF) demonstrates a significant advantage over EU methods (SISA, RecEraser, and UltraRE), particularly in terms of unlearning efficiency and recommendation fairness. However, it is essential to highlight that unlearning completeness is the primary goal of unlearning. EU methods inherently achieve the highest level of completeness, whereas SCIF can only achieve weak unlearning.

### 5.1 Unlearning Completeness

To evaluate the completeness of AU methods, we report the accuracy of MIO in Table 2, where the recommendation model is WMF. Due to the space limit, we report the results of other models in Appendix A.2. Compared the result of SCIF with the

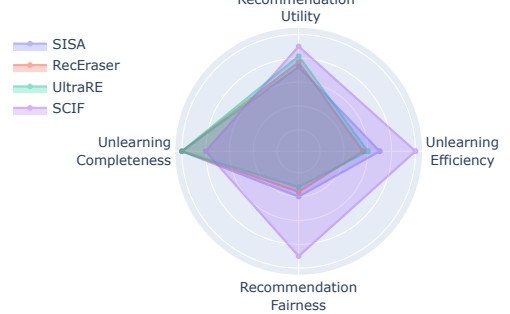

Figure 2: A visualized evaluation overview of recommendation unlearning methods in four aspects (↑), where the result is the normalized average outcome obtained across all models and datasets, using random data as the unlearning set. The recommendation fairness is measured by A-IGF (fairness between active and inactive users). The higher values represent better performance.

performance before unlearning and Retrain after unlearning, we observe that i) both SCIF and Retrain significantly decrease the MIO accuracy, indicating their effectiveness in unlearning; ii) although not significant, there is still a marginal gap between SCIF and Retrain (ground truth), i.e., 4.1% higher accuracy than Retrain on average; and iii) SCIF particularly performance worse on edge data compared to other data types. This discrepancy may be attributed to imprecise influence estimation for this specific data category.

### 5.2 Recommendation Utility

We report the results in terms of recommendation utility for AU and EU in Tables 2 and 3, respectively. In general, the AU method (SCIF) outperforms the EU methods (SISA, RecEraser, and UltraRE). Employing the same unlearning set, RecEraser and UltraRE consistently outperform SISA across all datasets and models, with UltraRE generally surpassing RecEraser, aligning with previous research (Li et al., 2023a).

Table 3: Results in terms of recommendation utility for exact recommendation unlearning methods.

| ML-100K | | Retrain | | | SISA | | | RecEraser | | | UltraRE | | |
|---|---|---|---|---|---|---|---|---|---|---|---|---|---|
| | | Core | Random | Edge | Core | Random | Edge | Core | Random | Edge | Core | Random | Edge |
| WMF | NDCG@20 | 0.3187 | 0.2872 | 0.3091 | 0.2096 | 0.2092 | 0.2041 | 0.2285 | 0.2208 | 0.2109 | 0.2303 | 0.2354 | 0.2149 |
| | HR@20 | 0.3295 | 0.3353 | 0.3140 | 0.2094 | 0.2049 | 0.1892 | 0.2218 | 0.2142 | 0.1979 | 0.2267 | 0.2282 | 0.2027 |
| BPR | NDCG@20 | 0.3111 | 0.3003 | 0.3043 | 0.2244 | 0.2324 | 0.2298 | 0.2614 | 0.2615 | 0.2694 | 0.2708 | 0.2764 | 0.2743 |
| | HR@20 | 0.3151 | 0.3028 | 0.2987 | 0.2203 | 0.2259 | 0.2179 | 0.2724 | 0.2658 | 0.2620 | 0.2851 | 0.2813 | 0.2695 |
| LightGCN | NDCG@20 | 0.3175 | 0.3121 | 0.3101 | 0.1802 | 0.1932 | 0.1964 | 0.2856 | 0.2905 | 0.2886 | 0.2952 | 0.3069 | 0.3063 |
| | HR@20 | 0.3250 | 0.3253 | 0.3244 | 0.1724 | 0.1907 | 0.1911 | 0.3053 | 0.3099 | 0.3121 | 0.3123 | 0.3201 | 0.3185 |

| ML-1M | | Retrain | | | SISA | | | RecEraser | | | UltraRE | | |
|---|---|---|---|---|---|---|---|---|---|---|---|---|---|
| | | Core | Random | Edge | Core | Random | Edge | Core | Random | Edge | Core | Random | Edge |
| WMF | NDCG@20 | 0.2196 | 0.2124 | 0.2148 | 0.1780 | 0.1639 | 0.1714 | 0.1894 | 0.1796 | 0.1838 | 0.1926 | 0.1891 | 0.1970 |
| | HR@20 | 0.2174 | 0.2108 | 0.2051 | 0.1612 | 0.1485 | 0.1493 | 0.1731 | 0.1592 | 0.1596 | 0.1747 | 0.1680 | 0.1717 |
| BPR | NDCG@20 | 0.2462 | 0.2319 | 0.2336 | 0.1545 | 0.1530 | 0.1628 | 0.1826 | 0.1660 | 0.1860 | 0.1828 | 0.1856 | 0.1913 |
| | HR@20 | 0.2279 | 0.2162 | 0.2118 | 0.1353 | 0.1329 | 0.1367 | 0.1627 | 0.1450 | 0.1624 | 0.1652 | 0.1632 | 0.1651 |
| LightGCN | NDCG@20 | 0.2177 | 0.2108 | 0.2147 | 0.1504 | 0.1533 | 0.1642 | 0.1864 | 0.1863 | 0.1814 | 0.1969 | 0.1867 | 0.1806 |
| | HR@20 | 0.2138 | 0.2045 | 0.2186 | 0.1365 | 0.1323 | 0.1581 | 0.1825 | 0.1804 | 0.1818 | 0.1907 | 0.1855 | 0.1798 |

| ADM | | Retrain | | | SISA | | | RecEraser | | | UltraRE | | |
|---|---|---|---|---|---|---|---|---|---|---|---|---|---|
| | | Core | Random | Edge | Core | Random | Edge | Core | Random | Edge | Core | Random | Edge |
| WMF | NDCG@20 | 0.3691 | 0.3566 | 0.3556 | 0.2720 | 0.2589 | 0.2515 | 0.3373 | 0.3256 | 0.3185 | 0.3420 | 0.3334 | 0.3347 |
| | HR@20 | 0.4071 | 0.3822 | 0.3848 | 0.2617 | 0.2492 | 0.2471 | 0.3527 | 0.3467 | 0.3203 | 0.3689 | 0.3595 | 0.3501 |
| BPR | NDCG@20 | 0.3566 | 0.3453 | 0.3499 | 0.2806 | 0.2708 | 0.2757 | 0.3286 | 0.3295 | 0.3212 | 0.3325 | 0.3301 | 0.3314 |
| | HR@20 | 0.3821 | 0.3628 | 0.3718 | 0.2745 | 0.2638 | 0.2611 | 0.3486 | 0.3406 | 0.3483 | 0.3541 | 0.3569 | 0.3608 |
| LightGCN | NDCG@20 | 0.0105 | 0.0106 | 0.0096 | 0.0075 | 0.0054 | 0.0048 | 0.0084 | 0.0085 | 0.0079 | 0.0097 | 0.0088 | 0.0086 |
| | HR@20 | 0.0221 | 0.0234 | 0.0208 | 0.0157 | 0.0112 | 0.0103 | 0.0171 | 0.0176 | 0.0154 | 0.0191 | 0.0185 | 0.0183 |

Table 4: Results in terms of unlearning efficiency (running time in seconds ↓).

| Time (s) | | ML-100K | | | ML-1M | | | ADM | | |
|---|---|---|---|---|---|---|---|---|---|---|
| | | WMF | BPR | LightGCN | WMF | BPR | LightGCN | WMF | BPR | LightGCN |
| Retrain | Core | 4296 | 5238 | 4734 | 7748 | 9113 | 8645 | 3682 | 6998 | 5225 |
| | Random | 4526 | 5494 | 5044 | 8693 | 9461 | 10324 | 3972 | 7127 | 5354 |
| | Edge | 4687 | 5527 | 5274 | 8006 | 9748 | 10497 | 4127 | 7351 | 6359 |
| SISA | Core | 402 | 488 | 437 | 1160 | 1160 | 1523 | 669 | 1750 | 1009 |
| | Random | 467 | 586 | 528 | 1256 | 1265 | 1605 | 717 | 1842 | 1246 |
| | Edge | 442 | 504 | 515 | 1280 | 1292 | 1659 | 751 | 1902 | 1077 |
| RecEraser | Core | 463 | 582 | 561 | 1533 | 1568 | 1846 | 865 | 1892 | 1106 |
| | Random | 476 | 693 | 656 | 1654 | 1660 | 1952 | 912 | 1945 | 1490 |
| | Edge | 489 | 659 | 617 | 1736 | 1819 | 1964 | 965 | 2032 | 1190 |
| UltraRE | Core | 457 | 591 | 559 | 1507 | 1493 | 1667 | 819 | 1810 | 1057 |
| | Random | 482 | 618 | 645 | 1595 | 1550 | 1834 | 901 | 1862 | 1283 |
| | Edge | 466 | 518 | 666 | 1781 | 1791 | 1955 | 923 | 1904 | 1368 |
| SCIF | Core | 289 | 336 | 316 | 784 | 784 | 1034 | 453 | 1186 | 682 |
| | Random | 325 | 403 | 368 | 862 | 860 | 1083 | 497 | 1242 | 841 |
| | Edge | 316 | 358 | 359 | 887 | 877 | 1126 | 520 | 1282 | 733 |

For all EU methods, the recommendation utility of unlearning core users is generally higher than that of unlearning random-select or edge users. This is likely due to the removal of data from more interactive users, which typically contains a large amount of ratings. This enables the model to learn more effectively from the smaller amount of remaining training data. Compared with these EU methods, SCIF exhibits the highest recommendation utility, closely resembling that of Retrain. However, SCIF suffers the most substantial performance decline when unlearning core users. This can be attributed to the increased number of interactions involved in calculating the influence function, leading to inaccurate influence estimation that negatively impacts the model utility.

## 5.3 Unlearning Efficiency

We report the unlearning times in Table 4. In general, SCIF is more efficient than EU methods. Among the EU methods, SISA saves more time compared to RecEraser and UltraRE, because it does not have the complex division and aggregation stage specific to the recommendation scenarios. Due to its design, UltraRE is slightly more efficient than RecEraser. Additionally, EU methods take

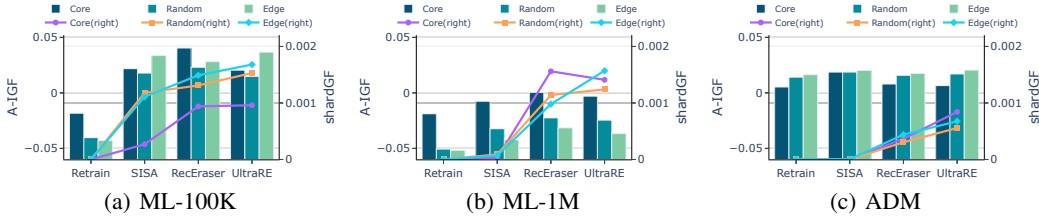

| (a) ML-100K | (b) ML-1M | (c) ADM |

Figure 3: Results in terms of recommendation fairness for exact recommendation unlearning methods on WMF, where A-IGF (approaching Retrain) and shardGF (↓) evaluate the fairness of group-level and shard-level, respectively.

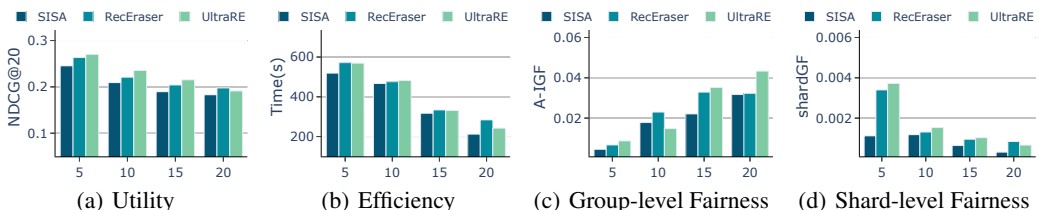

| (a) Utility | (b) Efficiency | (c) Group-level Fairness | (d) Shard-level Fairness |

Figure 4: Effect of shard number in terms of multiple aspects, i.e., recommendation utility (↑), unlearning efficiency (↓), group-level fairness (approaching Retrain), and shard-level fairness (↓).

less time to unlearn core users since they have a larger amount of interaction data. This reduces the amount of data left for retraining. On the contrary, SCIF requires more computations for influence estimation on core users, resulting in higher time costs compared to unlearning random or edge users.

### 5.4 Recommendation Fairness

We also report the recommendation fairness of AU and EU methods in Table 2 and Figure 3, respectively. For the *group-level fairness* (A-IGF), compared to the AU method (SCIF), EU methods notably worsen unfairness, tending to favor active users. This is primarily attributed to the division stage of EU methods, with this effect becoming more pronounced on larger datasets, i.e., ML-1M and ADM. Moreover, RecEraser and UltraRE, which group active users together instead of randomly, as done by SISA, exacerbate unfairness even further. For the *shard-level fairness* (shardGF), although to a lesser extent compared to group-level fairness, RecEraser and UltraRE also exacerbate unfairness.

### 5.5 Effects of Shard Number

We report the effect of shard number in terms of multiple aspects in Figure 4, using WMF on ML-100K. *Firstly*, as the number of shards increases, the unlearning efficiency improves, but the recommendation utility deteriorates, as confirmed by several previous studies (Chen et al., 2022; Li et al., 2023a). *Secondly*, the increased shard number further groups the active users into smaller shards, exacerbating the group-level fairness. At the same time, it reduces the discrepancy among all shards, diminishing the shard-level fairness.

## 6 Conclusion

In this paper, we present a comprehensive benchmark, CURE4Rec, for recommendation unlearning methods, aiming to analyze and inspire further exploration into the deeper influence of recommendation unlearning. Specifically, CURE4Rec covers four evaluation aspects, i.e., unlearning completeness, recommendation utility, unlearning efficiency, and recommendation fairness. Additionally, we investigate unlearning robustness across three unlearning sets, i.e., core data, edge data, and random data. Through extensive experiments, we compare the performance of various recommendation unlearning methods using our proposed benchmark. Our experiments reveal that i) the division-aggregation design of the EU approach has dual implications. On one hand, it inherently

achieves unlearning completeness. On the other hand, it compromises other evaluation aspects. and ii) The AU approach, which directly manipulates model parameters, outperforms the EU approach in all aspects except completeness, with less negative influence on model properties, e.g., fairness.

**Limitation and Boarder Impact.** This paper proposes a benchmark for recommendation unlearning, comprising four evaluation aspects. This design also offers insights for other unlearning scenarios. Simultaneously, there is considerable room for improvement in the specific evaluation metrics within each aspect. Additionally, the AU approach appears to outperform the EU approach in all aspects except completeness. The trade-off between completeness and other aspects is an intriguing direction that is not discussed in this paper.

## Acknowledgments and Disclosure of Funding

This work was supported by the Fundamental Research Funds for the Central Universities 226-2024-00241. We thank all the anonymous reviewers for helpful feedback on early versions of this work.

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

# A  More Results

## A.1  Performance Overview

We report a visualized overview of compared recommendation unlearning methods on each dataset in Figure 5. The results are generally consistent with Figure 2.

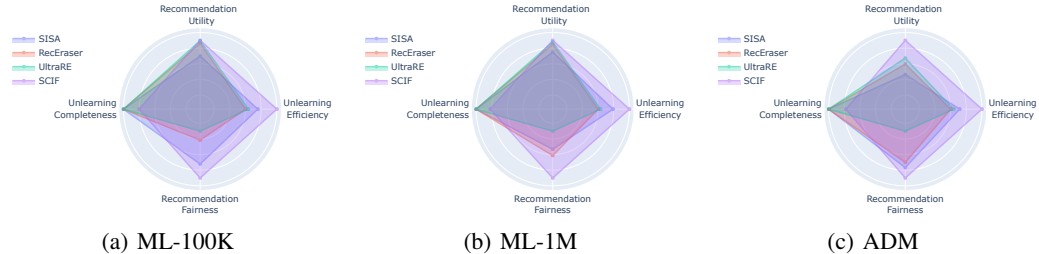

(a) ML-100K  (b) ML-1M  (c) ADM

Figure 5:  A visualized evaluation overview of recommendation unlearning methods in four aspects (↑), where the result is the normalized average outcome obtained across all models, using random data as the unlearning set. The recommendation fairness is measured by A-IGF (fairness between active and inactive users).

## A.2  Unlearning Completeness

We report the accuracy of MIO in Table 5, where the recommendation model is BPR. We omit the results for LightGCN as we encountered difficulties accurately computing the influence function of SCIF on ML-1M and ADM based on current hardware.

Table 5: Results in terms of unlearning completeness (MIO accuracy - approaching 0.5), recommendation utility (NDCG and HR ↑), and recommendation fairness (A-IGF - approaching Retrain) for the approximate recommendation unlearning method, where `Learn` denotes the results before unlearning. Core, random, and edge respectively refer to the selection of the unlearning sets as core data, random data, and edge data.

|  |  | ML-100K | | | | ML-1M | | | | ADM | | | |
|---|---|---|---|---|---|---|---|---|---|---|---|---|---|
|  |  | NDCG@20 | HR@20 | MIO | A-IGF | NDCG@20 | HR@20 | MIO | A-IGF | NDCG@20 | HR@20 | MIO | A-IGF |
| Learn | | 0.3195 | 0.3030 | 0.724 | -0.0246 | 0.2517 | 0.2306 | 0.744 | -0.0651 | 0.0251 | 0.0510 | 0.759 | 0.0194 |
| Retrain | Core | 0.3111 | 0.3151 | 0.536 | -0.0217 | 0.2462 | 0.2279 | 0.549 | -0.0374 | 0.0246 | 0.0504 | 0.558 | 0.0066 |
| | Random | 0.3003 | 0.3028 | 0.535 | -0.0153 | 0.2319 | 0.2162 | 0.550 | -0.0605 | 0.0203 | 0.0421 | 0.561 | 0.0187 |
| | Edge | 0.3043 | 0.2987 | 0.537 | -0.0175 | 0.2336 | 0.2118 | 0.552 | -0.0633 | 0.0203 | 0.0439 | 0.555 | 0.0191 |
| SCIF | Core | 0.2392 | 0.2182 | 0.565 | -0.0116 | 0.1898 | 0.1636 | 0.572 | -0.0284 | 0.0171 | 0.0336 | 0.573 | 0.0096 |
| | Random | 0.2768 | 0.2824 | 0.566 | -0.0144 | 0.2159 | 0.1886 | 0.576 | -0.0372 | 0.0189 | 0.0357 | 0.573 | 0.0110 |
| | Edge | 0.2871 | 0.2905 | 0.612 | -0.0167 | 0.2231 | 0.1942 | 0.635 | -0.0481 | 0.0200 | 0.0417 | 0.588 | 0.0132 |

## A.3  Recommendation Fairness

We report the recommendation fairness of exact unlearning methods on each dataset using BPR and LightGCN recommendation models in Figures 6 and 7, respectively.

We also report the grouping results of active and inactive users after applying three exact unlearning methods, i.e., SISA, RecEraser, UltraRE, on different datasets in Tables 6, 7, and 8. On the one hand, SISA randomly distributes both types of users evenly across groups. On the other hand, RecEraser and UltraRE tend to cluster active users into the same groups, which results in certain groups containing numerous active users while others have almost none. This clustering result explains why RecEraser and UltraRE tend to favor active users, as the concentration of active users in certain groups significantly increases their proportion compared to random distribution, leading to more effective learning but also more severe unfairness.

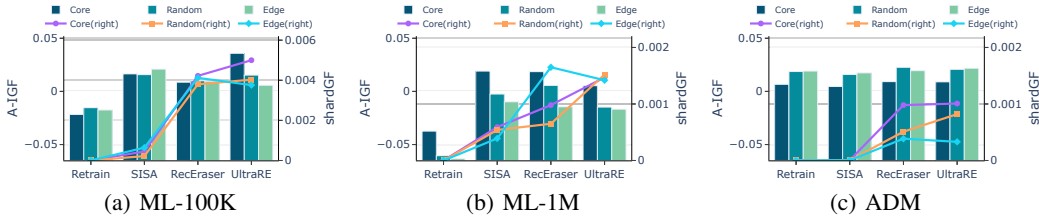

(a) ML-100K  (b) ML-1M  (c) ADM

Figure 6: Results in terms of recommendation fairness for exact recommendation unlearning methods on BPR, where A-IGF (approaching Retrain) and shardGF (↓) evaluate the fairness of group-level and shard-level, respectively.

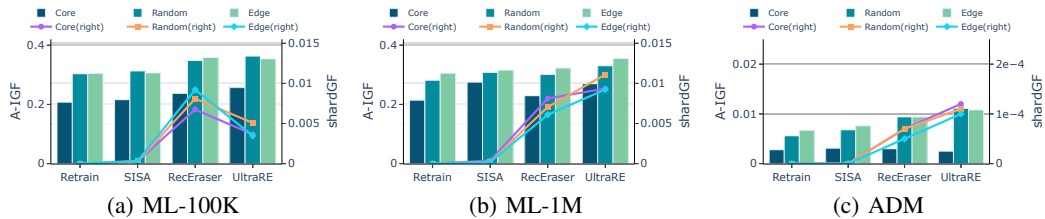

(a) ML-100K  (b) ML-1M  (c) ADM

Figure 7: Results in terms of recommendation fairness for exact recommendation unlearning methods on LightGCN, where A-IGF (approaching Retrain) and shardGF (↓) evaluate the fairness of group-level and shard-level, respectively.

## A.4  Unlearning Ratio

We report the effect of unlearning data ratio in terms of multiple aspects in Figure 8, using WMF on ML-100K. We observe consistent results with previous studies (Bourtoule et al., 2021; Chen et al., 2022; Li et al., 2023a). In general, as the ratio of unlearning data increases, the recommendation utility gradually decreases, along with a reduction in the unlearning time. Additionally, a larger unlearning ratio tends to lead to greater fairness.

Table 6: Results of user distribution (active vs. inactive) in each shard on dataset ML-100K. The unlearning data ratio is set to 5%.

| ML-100K | | Group 1 | | Group 2 | | Group 3 | | Group 4 | | Group 5 | |
|---|---|---|---|---|---|---|---|---|---|---|---|
| | | Active | Inactive | Active | Inactive | Active | Inactive | Active | Inactive | Active | Inactive |
| SISA | Core | 4 | 86 | 4 | 86 | 3 | 87 | 7 | 83 | 8 | 82 |
| | Random | 4 | 86 | 8 | 82 | 2 | 88 | 3 | 87 | 6 | 84 |
| | Edge | 3 | 87 | 6 | 84 | 3 | 87 | 6 | 84 | 6 | 84 |
| RecEraser | Core | 0 | 90 | 10 | 80 | 0 | 90 | 0 | 90 | 0 | 86 |
| | Random | 0 | 90 | 1 | 89 | 1 | 89 | 0 | 90 | 0 | 90 |
| | Edge | 0 | 90 | 6 | 84 | 0 | 90 | 0 | 86 | 1 | 89 |
| UltraRE | Core | 0 | 89 | 11 | 78 | 0 | 90 | 6 | 83 | 0 | 90 |
| | Random | 0 | 90 | 3 | 86 | 3 | 87 | 1 | 89 | 0 | 89 |
| | Edge | 0 | 89 | 15 | 75 | 1 | 88 | 2 | 87 | 7 | 83 |
| ML-100K | | Group 6 | | Group 7 | | Group 8 | | Group 9 | | Group 10 | |
| | | Active | Inactive | Active | Inactive | Active | Inactive | Active | Inactive | Active | Inactive |
| SISA | Core | 1 | 89 | 3 | 86 | 3 | 86 | 6 | 83 | 5 | 84 |
| | Random | 5 | 85 | 4 | 85 | 1 | 88 | 5 | 84 | 6 | 83 |
| | Edge | 8 | 82 | 2 | 87 | 1 | 88 | 7 | 82 | 2 | 87 |
| RecEraser | Core | 0 | 90 | 0 | 90 | 28 | 62 | 0 | 90 | 6 | 84 |
| | Random | 6 | 84 | 9 | 81 | 0 | 90 | 27 | 63 | 0 | 86 |
| | Edge | 0 | 90 | 9 | 81 | 0 | 90 | 27 | 63 | 1 | 89 |
| UltraRE | Core | 7 | 83 | 10 | 80 | 1 | 89 | 0 | 89 | 9 | 81 |
| | Random | 3 | 87 | 1 | 88 | 0 | 89 | 18 | 72 | 15 | 75 |
| | Edge | 0 | 90 | 1 | 89 | 0 | 90 | 7 | 82 | 11 | 79 |

Table 7: Results of user distribution (active vs. inactive) in each shard on dataset ML-1M. The unlearning data ratio is set to 5%.

| ML-1M | | Group 1 | | Group 2 | | Group 3 | | Group 4 | | Group 5 | |
|---|---|---|---|---|---|---|---|---|---|---|---|
| | | Active | Inactive | Active | Inactive | Active | Inactive | Active | Inactive | Active | Inactive |
| SISA | Core | 28 | 546 | 31 | 543 | 26 | 548 | 30 | 544 | 32 | 542 |
| | Random | 25 | 549 | 34 | 540 | 30 | 544 | 23 | 551 | 35 | 539 |
| | Edge | 36 | 538 | 20 | 554 | 24 | 550 | 32 | 542 | 27 | 547 |
| RecEraser | Core | 44 | 530 | 52 | 522 | 0 | 572 | 20 | 554 | 5 | 569 |
| | Random | 2 | 570 | 79 | 495 | 44 | 530 | 40 | 534 | 5 | 569 |
| | Edge | 10 | 564 | 41 | 533 | 74 | 500 | 31 | 543 | 0 | 574 |
| UltraRE | Core | 0 | 573 | 5 | 569 | 11 | 563 | 12 | 562 | 33 | 541 |
| | Random | 44 | 530 | 6 | 567 | 7 | 567 | 5 | 569 | 13 | 561 |
| | Edge | 8 | 566 | 9 | 564 | 11 | 563 | 4 | 569 | 23 | 550 |

| ML-1M | | group6 | | Group 7 | | Group 8 | | Group 9 | | Group 10 | |
|---|---|---|---|---|---|---|---|---|---|---|---|
| | | Active | Inactive | Active | Inactive | Active | Inactive | Active | Inactive | Active | Inactive |
| SISA | Core | 23 | 551 | 25 | 549 | 33 | 541 | 28 | 545 | 30 | 543 |
| | Random | 28 | 546 | 38 | 536 | 22 | 552 | 30 | 543 | 21 | 552 |
| | Edge | 27 | 547 | 39 | 535 | 33 | 541 | 26 | 547 | 22 | 551 |
| RecEraser | Core | 64 | 510 | 1 | 573 | 38 | 536 | 35 | 539 | 27 | 547 |
| | Random | 32 | 542 | 1 | 573 | 61 | 513 | 2 | 572 | 20 | 554 |
| | Edge | 24 | 550 | 2 | 572 | 91 | 483 | 4 | 568 | 9 | 565 |
| UltraRE | Core | 44 | 530 | 27 | 547 | 18 | 556 | 32 | 541 | 104 | 470 |
| | Random | 3 | 571 | 0 | 574 | 14 | 559 | 7 | 567 | 187 | 387 |
| | Edge | 0 | 575 | 49 | 525 | 26 | 548 | 155 | 419 | 1 | 573 |

Table 8: Results of user distribution (active vs inactive) in each shard on dataset ADM. The unlearning data ratio is set to 5%.

| ADM | | Group 1 | | Group 2 | | Group 3 | | Group 4 | | Group 5 | |
|---|---|---|---|---|---|---|---|---|---|---|---|
| | | Active | Inactive | Active | Inactive | Active | Inactive | Active | Inactive | Active | Inactive |
| SISA | Core | 96 | 2078 | 113 | 2061 | 126 | 2048 | 117 | 2057 | 106 | 2068 |
| | Random | 108 | 2066 | 112 | 2062 | 95 | 2079 | 110 | 2064 | 84 | 2090 |
| | Edge | 112 | 2062 | 105 | 2069 | 100 | 2074 | 120 | 2054 | 106 | 2068 |
| RecEraser | Core | 429 | 1745 | 0 | 2174 | 8 | 2166 | 0 | 2174 | 0 | 2169 |
| | Random | 0 | 2169 | 0 | 2174 | 453 | 1721 | 159 | 2015 | 84 | 2090 |
| | Edge | 149 | 2025 | 84 | 2090 | 379 | 1795 | 7 | 2167 | 0 | 2169 |
| UltraRE | Core | 91 | 2083 | 160 | 2013 | 65 | 2108 | 65 | 2109 | 88 | 2086 |
| | Random | 41 | 2132 | 80 | 2094 | 361 | 1813 | 81 | 2093 | 56 | 2117 |
| | Edge | 82 | 2092 | 11 | 2162 | 201 | 1972 | 53 | 2120 | 330 | 1844 |

| ADM | | Group 6 | | Group 7 | | Group 8 | | Group 9 | | Group 10 | |
|---|---|---|---|---|---|---|---|---|---|---|---|
| | | Active | Inactive | Active | Inactive | Active | Inactive | Active | Inactive | Active | Inactive |
| SISA | Core | 98 | 2075 | 112 | 2061 | 99 | 2074 | 115 | 2058 | 104 | 2069 |
| | Random | 119 | 2054 | 114 | 2059 | 135 | 2038 | 116 | 2057 | 93 | 2080 |
| | Edge | 109 | 2064 | 106 | 2067 | 111 | 2062 | 102 | 2071 | 115 | 2058 |
| RecEraser | Core | 1 | 2173 | 97 | 2077 | 388 | 1786 | 121 | 2053 | 42 | 2132 |
| | Random | 9 | 2165 | 0 | 2174 | 0 | 2174 | 4 | 2170 | 377 | 1797 |
| | Edge | 456 | 1718 | 0 | 2174 | 0 | 2174 | 10 | 2164 | 1 | 2173 |
| UltraRE | Core | 65 | 2109 | 173 | 2001 | 147 | 2026 | 123 | 2051 | 109 | 2063 |
| | Random | 200 | 1973 | 50 | 2124 | 137 | 2036 | 48 | 2125 | 32 | 2142 |
| | Edge | 82 | 2091 | 58 | 2116 | 121 | 2052 | 55 | 2119 | 93 | 2081 |

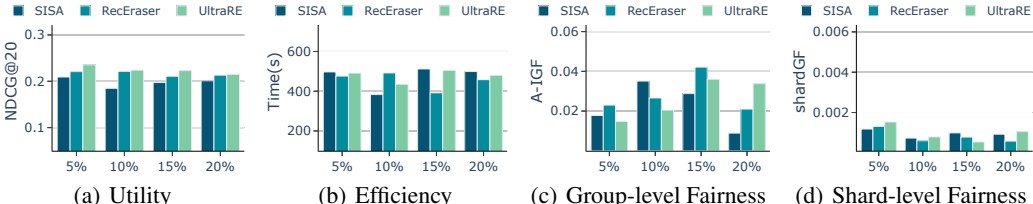

(a) Utility     (b) Efficiency     (c) Group-level Fairness     (d) Shard-level Fairness

Figure 8: Effect of unlearning ratio in terms of multiple aspects, i.e., recommendation utility ($\uparrow$), unlearning efficiency ($\downarrow$), group-level fairness (approaching Retrain), and shard-level fairness ($\downarrow$).

