# OpenReview forum: "CURE4Rec: A Benchmark for Recommendation Unlearning with Deeper Influence"
_NeurIPS.cc/2024/Datasets_and_Benchmarks_Track — NeurIPS 2024 Track Datasets and Benchmarks Poster_

### Official Review · Reviewer_YV6m · 2024-07-11
**Comments**

**Rating:** 8
**Confidence:** 4
**Correctness:** The evaluation methods and experiment…
**Clarity:** The paper is generally well-written.

**Review:**

This paper proposes a benchmark for evaluating the impact of machine unlearning methods on recommender systems.

**pros:**

"Unlearning" is a highly popular research direction at present, with a substantial amount of related research across various fields. It also holds significant potential in the realm of recommender systems. This paper can advance research in this novel direction. This paper explores the deeper impacts of machine unlearning on recommender system, particularly analyzing the performance of different unlearning methods under various data selection strategies. It also examines the impact of unlearning methods on the original fairness of recommender systems. This provides a foundation for future research on more robust and fair unlearning methods.

**cons:**

1. Additional fair retraining method.
2. Necessity of unlearning data selection strategies.
3. Fairness metrics.
4. Paper version.

**Strengths:**

"Unlearning" is a highly popular research direction at present, with a substantial amount of related research across various fields. It also holds significant potential in the realm of recommender systems. This paper can advance research in this novel direction. This paper explores the deeper impacts of machine unlearning on recommender system, particularly analyzing the performance of different unlearning methods under various data selection strategies. It also examines the impact of unlearning methods on the original fairness of recommender systems. This provides a foundation for future research on more robust and fair unlearning methods.

**Additional Feedback:**

None.

**Documentation:**

The benchmark code is provided.

**Ethics:**

No ethical concerns.

**Limitations:**

The paper clearly states the limitation of the specific evaluation metrics in each aspect.

**Opportunities For Improvement:**

1. Additional fair retraining method: The authors explored the impact of different unlearning methods on recommendation models, but these models are based on traditional training methods, which are not designed with fairness as a primary goal. To more clearly investigate the impact of unlearning on fairness, unlearning should be conducted on a fair recommendation model, and various metrics should be evaluated.
2. Necessity of unlearning data selection strategies: The paper presents the impact of core-user, random-user, and edge-user unlearning strategies. However, aside from a clear pattern in unlearning time, these three strategies have little impact on other aspects.
3. Fairness metrics: The authors used two fairness metrics to analyze two types of unfairness. However, these metrics are based on recommendation performance, and different unlearning methods inherently result in different recommendation performances. Changes in fairness metrics may not truly reflect the impact of unlearning methods on model fairness but rather changes brought by recommendation performance.
4. Paper version: There is a discrepancy between the paper version in the system and the one in the supplementary materials.

**Relation To Prior Work:**

The related work is well discussed.

**Summary And Contributions:**

This paper proposes a benchmark for evaluating the impact of machine unlearning methods on recommender systems. The paper examines the performance of five existing unlearning methods, including retraining, in terms of recommendation effectiveness, unlearning effectiveness, unlearning efficiency, and fairness. The paper also introduces an unlearn data selection method, providing three types of unlearn data, to investigate robustness. The study utilizes three datasets and three recommendation models. Additionally, the authors have open-sourced the benchmark code.

---

> ### Author Rebuttal · Authors · 2024-08-11
>
> **1. Additional fair retraining method**
> > **Response**: Indeed, fair approach can be adapted for retraining. However, it is not applicable to other unlearning methods. Specifically, existing fair recommendation methods can be primarily categorized into three types: pre-processing, in-processing, and post-processing [1]. Firstly, the grouping methods used in pre-processing approaches conflict with the grouping strategies in exact unlearning methods, making them inapplicable. Secondly, in-processing and post-processing approaches focus on global fairness, which may not be effective for exact unlearning. This is because exact unlearning divides dataset into multiple independent shard, breaking the global dataset into local segments. On the other hand, approximate unlearning alters the learning process, making fair training methods inapplicable.
>
> [1] Wang, Y., Ma, W., Zhang, M., Liu, Y., & Ma, S. (2023). A survey on the fairness of recommender systems. ACM Transactions on Information Systems, 41(3), 1-43.
>
>
> **2. Necessity of unlearning data selection strategies**
> > **Response**: Data selection strategy is necessary, and our empirical results show performance differences among various selected users. In addition to unlearning efficiency, different users exhibit variations in model utility. Specifically, the utility for core users is generally higher compared to the other two user types.
>
> **3. Fairness metrics**
> > **Response**: Firstly, what we aim to evaluate is user-oriented fairness, which belongs to performance fairness. Thus, we use performance metrics for evaluation. Secondly, as shown in Figure 2, even when the recommendation performance is similar, there are still significant differences in fairness caused by different unlearning methods. Thus, the performance metrics can reflect the impact of unlearning methods on fairness.
>
> **4. Paper version**
> > **Response**: We sincerely apologize for not providing a consistent PDF version. There are indeed slight differences between the two, and we will correct this error during revision.

---

> > ### Comment · Reviewer_YV6m · 2024-08-13
> >
> > I appreciate the authors' response, which addresses my concerns. Thus, I will increase the rating by 1. I recommend that the authors incorporate the changes discussed in the rebuttal into the final revision.

---

### Official Review · Reviewer_Cg64 · 2024-07-11
**CURE4Rec: A Benchmark for Recommendation Unlearning with Deeper Influence**

**Rating:** 7
**Confidence:** 4
**Correctness:** 1. This paper use recommendation perf…

**Review:**

This paper focus on an interesting topic of machine unlearning, particularly within the context of recommender systems. The proposed benchmark not only uses commonly-used evaluation metrics, but also investigate the deeper influence of unlearning, i.e., fairness. There are some issues in experiments and documentation, but I think it is easy to fix. Please refer to the following comments for details of pros and cons.

**Strengths:**

1. The topic of unlearning is gaining increasing popularity and appeals to a wide range of audiences. Although unlearning has not yet seen widespread industrial application, the recommender system focused on in this paper presents a practical application scenario.
2. This paper evaluates a side effect of unlearning (fairness), which is indeed insightful.
3. Some interesting findings are presented, demonstrating the advantages of approximate unlearning. This offers guidance for future directions.
4. The paper is well-written.

**Additional Feedback:**

NA

**Clarity:**

In general, this paper is well-organized and clearly written. However, the main content appears to be incomplete, different from the main content in appendix. Based on my understanding, the main content in the appendix should be the final version.

**Documentation:**

Code and example dataset are provided. But adding more detailed instructions on how to use them would be helpful.

**Ethics:**

Although this paper investigates the fairness of unlearning algorithms, there is no ethical concern in the proposed benchmark. The experiments were conducted on widely-used public datasets.

**Limitations:**

The authors mentioned that the absence of completeness evaluation on exact unlearning is a limitation. So why not conduct experiments on this? The proposed MIA-based evaluation is also applicable to exact unlearning.

**Opportunities For Improvement:**

Please refer to limitations, correctness, clarity, relation to prior work, and documentation.

**Relation To Prior Work:**

This paper discusses prior work in image classification, LLM, and diffusion. It focuses on recommendation tasks which is different from prior work. Another work on LLM is ignored [1].

[1] https://arxiv.org/abs/2406.10890

**Summary And Contributions:**

This paper proposes the first benchmark on recommendation unlearning, which consist of four aspects, including unlearning completeness, recommendation utility, unlearning efficiency, and recommendation fairness. Experiments were conducted on three datasets across three models, and code is provided.

---

> ### Author Rebuttal · Authors · 2024-08-11
>
> **1. Why not use metrics like statistical parity or equal opportunity**
> > **Response**: We apologize for this typo. What we aim to evaluate is user-oriented fairness [1,2], which belongs to performance fairness. Thus, we use performance metrics for evaluation. We will make corrections during the revision to prevent misunderstanding.
>
> [1] Li, Y., Chen, H., Fu, Z., Ge, Y., & Zhang, Y. (2021, April). User-oriented fairness in recommendation. In Proceedings of the web conference 2021 (pp. 624-632).
>
> [2] Han, Z., Chen, C., Zheng, X., Li, M., Liu, W., Yao, B., ... & Yin, J. (2024, March). Intra-and Inter-group Optimal Transport for User-Oriented Fairness in Recommender Systems. In Proceedings of the AAAI Conference on Artificial Intelligence (Vol. 38, No. 8, pp. 8463-8471).
>
> **2. Evaluating completeness of exact unlearning**
> > **Response**: Evaluating the completeness of exact unlearning will indeed provide valuable insights. We did not evaluate exact unlearning for the following reasons. Firstly, the current MIA-based evaluation method, while applicable, is inappropriate for exact unlearning because there are no negative samples for shadow training in this context. Using negative samples from approximate unlearning to evaluate exact unlearning is not suitable. Secondly, exact unlearning inherently achieves full authorized completeness. We will explore effective completeness evaluation methods for exact unlearning in future work.
>
> **3. Paper version**
> > **Response**: We sincerely apologize for not providing a consistent PDF version. There are indeed slight differences between the two, and we will correct this error during revision.
>
> **4. Additional related work**
> > **Response**: We apologize for the incomplete literature review. We will include the mentioned references in the revised version.
>
> **5. More details in github**
> > **Response**: Thank you for your suggestion. We have enhanced the README file in the GitHub repository by including detailed hyperparameter settings, usage instructions, and other relevant information.

---

### Official Review · Reviewer_UvtH · 2024-07-15
**Benchmark for evaluating existing machine unlearning methods**

**Rating:** 7
**Confidence:** 3
**Correctness:** The evaluation methods and experiment…
**Clarity:** The paper is well written.

**Review:**

Pros:

1. This paper propose a benchmark named CURE4Rec to evaluate the recommendation unlearning methods. Compared with existing work, the CURE4Rec firstly introduce fairness and robustness as two original metrics, which provide a new perspective to comprehensively review previous work.

2. Instead of directly utilizing the widely-used recommendation datasets, this paper proposes a selection strategy to create the unlearning set. According to this, the robustness of unlearning method is effectively evaluated.

3. The authors incorporated 5 baselines, which is great in understanding each method's performance in terms of robustness and fairness.

4. This paper is well-written and easy to read. The figures and tables offer a thorough representation of the results.

5. The conclusions in the paper offer possible directions for future work.

cons:
1.	More details are needed in github repository.

2.	Some related work needs to be cited.

See Opportunities For Improvement for details.

**Strengths:**

See Pros

**Additional Feedback:**

N/A

**Documentation:**

The URL of codes in abstract is accessible. But more details are needed.

**Ethics:**

No ethical concerns.

**Limitations:**

The paper discusses limitations.

**Opportunities For Improvement:**

1.	The content of Readme file in the github repository should be more detailed, including hyperparameter settings, usage instructions, etc.
2.	Some related work about LLM-based recommendation unlearning, such as [1] [2], should be cited in this paper.

Reference:

[1] Exact and Efficient Unlearning for Large Language Model-based Recommendation

[2] Towards Efficient and Effective Unlearning of Large Language Models for Recommendation

**Relation To Prior Work:**

See Opportunities For Improvement

**Summary And Contributions:**

In this paper, the authors propose a benchmark for evaluating existing machine unlearning methods under the scenario of recommender systems. Specifically, the proposed CURE4Rec introduce five metrics，including recommendation effectiveness, unlearning utility, unlearning efficiency, robustness and fairness, which establish a comprehensive framework for revisiting recommendation unlearning task. Furthermore, three public datasets and three recommendation models are used to conduct experiments. From the evaluation results, this study finds that all unlearning methods exhibit significant differences in utility and efficiency across different unlearning sets.

---

> ### Author Rebuttal · Authors · 2024-08-11
>
> **1. More details in github**
> > **Response**: Thank you for your suggestion. We have enhanced the README file in the GitHub repository by including detailed hyperparameter settings, usage instructions, and other relevant information.
>
> **2. Additional related work**
> > **Response**: We apologize for the incomplete literature review. We will include the mentioned references in the revised version. We will also include related work on model-specific recommendation unlearning, e.g., sequential recommendation and session-based recommendation.

---

### Official Review · Reviewer_nZM1 · 2024-07-17
**The authors design a comprehensive benchmark to evaluate recommendation unlearning methods. They evaluate recommendation unlearning methods from unlearning completeness, recommendation utility, unlearning efficiency and recommendation fairness. They conduct experiments with existing recommendation unlearning methods on multiple datasets and report their performance.**

**Rating:** 7
**Confidence:** 4

**Review:**

This paper proposes an interesting process to comprehensively evaluate recommendation unlearning methods. They fulfill a unified evaluation framework and consider the overlooked aspects of deeper influence of recommendation unlearning methods.

Pros:

1. The paper provides a comprehensive benchmark tailored for evaluating recommendation unlearning methods from four aspects.

2. The paper is well organized.  The evaluation framework and experiments are clear and easy to read.

Cons:

1. The first concern is whether evaluating fairness on different shards is reasonable. It appears that each shard has a distinct unlearning task, which affects the accuracy of each shard differently. It remains to be seen whether the unfairness observed in the unlearning model is caused by the degree of forgetting.

2. The authors mention that they evaluate the group fairness of recommendation unlearning in Section 3.2. However, it appears that the evaluation metrics used in the experiments pertain to performance fairness metrics. I think more explanation is warranted here.

**Strengths:**

* This paper provides a comprehensive benchmark tailored for evaluating recommendation unlearning methods from four aspects. This interesting field attracted the attention of researchers recently.
* The authors construct their evaluation system from multiple aspects.
* The authors evaluate several recommendation unlearning methods on all datasets. The experiments are well-developed and easy to understand.

**Additional Feedback:**

NA

**Clarity:**

Above average.

The paper is well organized.  The evaluation framework and experiments are clear and easy to read.

**Correctness:**

This paper proposes a comprehensive benchmark for a novel research problem. The evaluation methods and experiments are likely designed appropriately and performed correctly.

A minor comment: I think line 154 should be “we evaluate the performance fairness of recommendation unlearning” not "group fairness".

**Documentation:**

Yes. Well documented on Github.

**Ethics:**

No ethical concerns.

**Limitations:**

The authors are suggested to explain the rationale for evaluating fairness on different shards. Different forgetting tasks have varying effects on the accuracy of each shard. More evidence is needed to demonstrate that tasks of varying forgetting difficulty in each shard are not related to the observed unfairness phenomenon.

**Opportunities For Improvement:**

Please see cons.

The authors evaluate robustness of recommendation unlearning methods by conducting experiments on edge data, selected data and random data. More common attack techniques, such as label noise and poisoning attack, could be implemented to further evaluate the robustness of existing methods.

**Relation To Prior Work:**

Yes. The authors propose a comprehensive benchmark for a novel research problem.

**Summary And Contributions:**

The authors design a comprehensive benchmark to evaluate recommendation unlearning methods. They evaluate recommendation unlearning methods from unlearning completeness, recommendation utility, unlearning efficiency and recommendation fairness. They conduct experiments with existing recommendation unlearning methods on multiple datasets and report their performance.

---

> ### Author Rebuttal · Authors · 2024-08-11
>
> **1. Whether evaluating fairness on different shards is reasonable**
> > **Response**: Thank you for your observation and for raising this insightful point. Indeed, different shard causes varying degrees of performance degradation, which inevitably manifests as unfairness. Our aim is to evaluate performance discrepancies among different shards due to unlearning through a lens of user-oriented fairness. The diverse impacts of shard exactly highlight the necessity of evaluating performance fairness.
>
> **2. The used metrics pertain to performance fairness, not group fairness**
> > **Response**: Thank you for pointing out this typo. In this paper, we aim to evaluate the user-oriented fairness [1,2], which belongs to performance fairness. We will make corrections during the revision to prevent misunderstanding.
>
> [1] Li, Y., Chen, H., Fu, Z., Ge, Y., & Zhang, Y. (2021, April). User-oriented fairness in recommendation. In Proceedings of the web conference 2021 (pp. 624-632).
>
> [2] Han, Z., Chen, C., Zheng, X., Li, M., Liu, W., Yao, B., ... & Yin, J. (2024, March). Intra-and Inter-group Optimal Transport for User-Oriented Fairness in Recommender Systems. In Proceedings of the AAAI Conference on Artificial Intelligence (Vol. 38, No. 8, pp. 8463-8471).
>
> **3. Implement attack techniques for evaluation**
> > **Response**: Thank you for your suggestion. Conducting Attacks is indeed a meaningful way to evaluate robustness. However, this paper focuses on implicit feedback, where the labels are binary (0/1), making label noise and poisoning attacks nearly impossible to conduct. We will explore feasible attacks in our future work.

---

### Author Rebuttal · Authors · 2024-08-11

We sincerely thank all the reviewers for their valuable comments and suggestions, which are crucial for improving our work. We hope our response addresses your concerns.

---

### Decision · Program_Chairs · 2024-09-26

**Decision:**

Accept (Poster)

**Comment:**

The reviewers unanimously found the paper to be of high quality and significance for acceptance.